# Validation of the FACETS-OF-PPC as an Outcome Measure for Children with Severe Neurological Impairment and Their Families—A Multicenter Prospective Longitudinal Study

**DOI:** 10.3390/children8100905

**Published:** 2021-10-11

**Authors:** Sophie Pelke, Julia Wager, Benedikt B. Claus, Kathrin Stening, Boris Zernikow, Mandira Reuther

**Affiliations:** 1Pediatric Palliative Care Center Datteln, Children’s and Adolescents’ Hospital Datteln, 45711 Datteln, Germany; j.wager@pedscience.de (J.W.); b.claus@pedscience.de (B.B.C.); k.stening@kinderpalliativzentrum.de (K.S.); b.zernikow@kinderklinik-datteln.de (B.Z.); m.reuther@kinderpalliativzentrum.de (M.R.); 2Department of Children’s Pain Therapy and Paediatric Palliative Care, Faculty of Health, School of Medicine, Witten/Herdecke University, 58448 Witten, Germany; 3PedScience Research Institute, 45711 Datteln, Germany

**Keywords:** palliative care, pediatric, patient-centered outcome measures, factor analysis, multicenter study, prospective study, validation

## Abstract

Outcome measurement in pediatric palliative care (PPC) is receiving increasing attention. The FACETS-OF-PPC, a multidimensional outcome measure for children with severe neurological impairment, has been developed and partly validated. This study aimed to conclude the validity of the German version of the FACETS-OF-PPC. A multicenter prospective study with two points of measurement has been conducted, employing confirmatory factor analyses, reliability analyses, and analyses to evaluate the tool’s sensitivity to change. Overall, 25 inpatient and outpatient teams throughout Germany recruited *N* = 227 parents of affected children and *N* = 238 professional caregivers. Participants filled out the FACETS-OF-PPC on the admission of a child to a palliative care service and at discharge from inpatient settings or two months after admission to outpatient services. The analyses revealed the questionnaire needing further adaption. Now, 17 of the original 34 items contribute to the construction of the questionnaire scales. The other items remain part of the questionnaire and may be evaluated descriptively. Furthermore, the FACETS-OF-PPC has moderate to appropriate internal consistency and is sensitive to change. Creating an outcome measure with good psychometric properties for the vulnerable population of children with severe neurological impairment appears extremely difficult. Considering these challenges, the FACETS-OF-PPC demonstrates adequate psychometric properties.

## 1. Introduction

Outcome measures in pediatric palliative care (PPC) should be multidimensional in order to capture its holistic nature, including physical, psychological, social, and spiritual aspects [1]. Until recently, no research on multidimensional outcome measurement focusing on pediatric palliative care patients with congenital and neurological conditions has been conducted. Even though these children constitute the largest patient group within pediatric palliative care in children under the age of 16 years [2], studies on multidimensional outcome measurement in this field have so far mainly focused on children with cancer [3,4]. However, recent research suggests that assuming the adequacy of outcome measures developed for pediatric cancer patients would be highly inappropriate for use in children with congenital and neurological conditions, thus emphasizing the need for targeted research [5]. While there exists one multidimensional outcome measure for pediatric palliative care, i.e., the APCA Children’s Palliative Outcome Scale [6], it is unclear whether this outcome, grounded on the expert consensus of African local experts, is applicable to other regions, and a validation study in Belgium did not consider children with severe neurological impairments [7].

A study by Pelke et al. [8] has developed a specific multidimensional outcome tool for children with congenital and neurological disorders who are affected by severe neurological impairment and their families. It is called the Family-Centered Multidimensional Outcome Measure for Pediatric Palliative Care, the FACETS-OF-PPC in short. The FACETS-OF-PPC takes into account the entire unit of care, meaning the child and its family. The first steps towards its validation in Germany have been taken in the conducted study [8]. However, certain steps are still missing in order to further substantiate the assessment of the tool and its adequacy as an outcome measure.

In the EAPC White Paper on outcome measurement in palliative care [1], Bausewein et al. point out that a tool’s psychometric properties, its validity and reliability, in particular, must be considered when judging its quality. The current study thus aims to provide the last steps in the assessment of the FACETS-OF-PPC’s validation in Germany in order to provide the first high-quality outcome measure for pediatric palliative care. We examine its factorial validity by investigating whether the factor structure found by Pelke et al. [8] can be replicated in a new independent sample. Furthermore, we investigate the reliability of the questionnaire scales and the tool’s ability to measure change in the assessed domains. 

## 2. Materials and Methods

### 2.1. Design

A multicenter prospective study was conducted to collect data for the assessment of the German version of the FACETS-OF-PPC. 

### 2.2. Setting

A total of 25 pediatric palliative care institutions throughout Germany aided in participant recruitment. Inpatient and outpatient institutions were included in the study in order to evaluate the outcome tool independently of the care setting (*n* = 1 pediatric palliative care unit, *n* = 9 children’s and adolescents’ hospices, and *n* = 15 pediatric palliative home care teams). 

### 2.3. Participants

Eligible for study participation were German-speaking parents of non-verbal children aged 0–25 years with palliative care needs and severe neurological impairment who were newly admitted to one of the palliative care institutions. Informed consent was mandatory. An acute crisis due to the child’s health status constituted the only exclusion criterion. Professional caregivers were eligible for study participation if they cared for children with the above-mentioned characteristics and if parents provided their consent. 

### 2.4. Recruitment and Data Collection

Data were collected from December 2019 to October 2020. A professional caregiver in charge of study supervision in the respective institution approached eligible families and verbally informed them about the study. If families were interested in participating, they received all study material in an unsealed prepaid return envelope when the child was admitted to the palliative care service and again at discharge for inpatient services and two months after admission for outpatient services. The differing time points for pre and post assessment in the two patient groups were chosen because the post assessment was used to measure sensitivity to change. As inpatient treatment is much more intense than outpatient treatment, which often consists of one weekly appointment only, the post assessment for outpatients was established at a later time point. Parents were asked to complete pre- and post-intervention assessments in inpatient as well as outpatient setting. The two samples (inpatient and outpatient) are not overlapping. Professional caregivers received all study materials after parents consented to their participation in writing. They also received all materials in a prepaid return envelope at the above-mentioned times. After filling out the study documents, participants sealed them in the envelope and sent them to the study coordinator (S.R.). No costs were incurred for them. 

### 2.5. Measures

The FACETS-OF-PPC (see Appendix A) consists of 39 items; 34 items are assigned to one of the 6 scales, namely, “symptoms”, “child’s social participation”, “normalcy”, “social support”, “coping with the disease”, and “caregiver’s competencies”. The remaining five items focus on additional symptoms, the parent’s partner as well as the ill child’s siblings, and they are to be evaluated descriptively. Most items are rated on a 6-point Likert scale ranging from 1 (completely disagree) to 6 (completely agree), mostly for the timeframe of the last seven days (see Appendix A). Symptoms are rated from 1 (not present) to 6 (very pronounced). A parent and a professional caregiver version of the questionnaire were used.

Four global ratings regarding the child’s symptoms, the child’s overall health status, the child’s quality of life, and the caregiver’s quality of life were assessed in order to be able to compare the tool’s scale scores against a general estimation of the concepts. Additionally, an evaluation questionnaire was employed, which assessed the acceptability of the questionnaire with regard to its length, comprehensibility, and relevance of items. Lastly, participants filled out a demographic questionnaire assessing information on the study participant (parents or professional caregiver), such as their relation to the child, their age and nationalities, in addition to information about the child, such as its age, sex, nationality, diagnosis, and the duration of receiving palliative care.

### 2.6. Analysis

Descriptive analyses were conducted in IBM SPSS 27.0. A confirmatory factor analysis was conducted separately for parent and professional caregiver data for both time points in R with the package lavaan [9] in order to assess the factorial validity of the FACETS-OF-PPC. The five items regarding siblings, partners, and additional symptoms included in the questionnaire are to be evaluated descriptively and have thus been excluded from the factor analysis. The model fit was interpreted according to the recommendations provided by Schreiber [10]: χ^2^/*df* (<3 = acceptable, <2 = good), Comparative Fit Index (CFI: >0.95 = acceptable), Tucker-Lewis Index (TLI: ≥0.95 = good), Standardized Root Mean Square Residual (SRMR ≤0.08 = good), and Root Mean Square Error of Approximation (RMSEA, <0.08 = acceptable; <0.05 = good). 

For both time points and questionnaire versions, McDonald’s *ω* was calculated to investigate the internal consistency (McDonald’s *ω* of between 0.7 and 0.9 is desirable), as the commonly used coefficient *α* (“Cronbach’s Alpha”) has the strict assumption of *τ*-equivalency [11]. The FACETS-OF-PPC’s sensitivity to change was determined for the parent and professional caregiver version by employing repeated-measures ANOVAs with false discovery rate correction [12]. This analysis is necessary because only outcome measures sensitive to change are useful in intervention or experimental research [13]. This is especially true when treatment effects are expected to be low, and samples are limited [14]. A sensitive outcome measure can furthermore be employed to differentiate between desired and undesired intervention effects [15].

## 3. Results

A total of *N* = 227 parents of *N* = 227 children and *N* = 238 professional caregivers were recruited for study participation (see Table 1a,b) at admission. Of these, *N* = 168 parents and *N* = 190 professional caregivers also participated at discharge or two months after admission to an outpatient service, respectively. The mean amount of time between the two time points of participation was 8.9 days for parents of children in inpatient settings and 74.5 days for parents of children in outpatient settings. For professional caregivers, the mean amount of time between the two time points of participation was 10.1 days for inpatient settings and 65.8 days for outpatient settings. 

Parents and professional caregivers rated the FACETS-OF-PPC in its current form to be adequate in length, well comprehensible, and encompassing relevant items (see Table 2). 

The confirmatory factor analysis of parent and professional caregiver data yielded unsatisfactory model fits (see Table 3—“original model”) and could thus not demonstrate factorial validity. We, therefore, tried to achieve the best model fit by refining the composition of the scales through deletion of items.

First, we reasoned that there might not be one common underlying factor influencing all of the assessed symptoms equally. We thus decided to eliminate “symptoms” as a scale of the FACETS-OF-PPC and to instead evaluate the indicated symptoms descriptively. This improved the model fit to that displayed in Table 3 under “revised model 1”. 

As the model fit was still insufficient, we continued to refine the scales by deleting individual items from the various scales. The deletion of items was conducted iteratively, always deleting the item with the worst loading until the deletion of items did not improve the model fit further (see Appendix A). Eventually, the model fit, as displayed in Table 3 under “revised model 2”, resulted after the deletion of additional nine items. All remaining 17 items constituting the questionnaire’s five scales are shown in Table 4. These should be used to calculate the respective scale scores. McDonald’s ω showed the scales’ internal consistencies ranged between 0.52 and 0.89 (see Table 5). 

Next, we calculated the scale means. These were used to assess the questionnaires sensitivity to change by calculating separate repeated measures ANOVAs for the parent and professional caregiver data. Results after false discovery rate correction for multiple comparisons show significant improvements on the scales “normalcy” (*F*(1, 92) = 70.5, *p* < 0.001) and “social support” (*F*(1, 92) = 42.9, *p* < 0.001) for parent data and on “child’s social participation” (*F*(1, 108) = 10.9, *p* < 0.01), “normalcy” (*F*(1, 105) = 59.4, *p* < 0.001), and “social support” (*F*(1, 103) = 46.7, *p* < 0.001) for professional caregiver data. 

When analyzing the indicated symptoms individually, repeated measures ANOVAs showed that significant improvements in symptom severity could be demonstrated for parent ratings of sleep (*F*(1, 76) = 12.1, *p*_adj_ = 0.007). As Figure 1 shows, parents and professional caregivers also rate the other symptoms as improved at discharge and two months after admission to an outpatient setting. However, the changes between their first and their second rating do not reach statistical significance (see Table 6). Furthermore, as the symptom scale has been removed from the questionnaire, we examined whether calculating the sum or the mean of indicated symptom items best represents the global rating of the child’s current health status as indicated by the parents and professional caregivers. For this, we correlated the sum or mean of symptom items with the global rating of the child’s current health status. Calculating the mean represented the child’s health status the best (for parent data: *r* = 0.30, *p* < 0.001; for professional caregiver data: *r* = 0.56, *p* < 0.001). A significant improvement of the symptom mean was found between the two points of measurement for parents (*F*(1, 152) = 12.8, *p* < 0.001) as well as professional caregivers (*F*(1, 174) = 8.33, *p* < 0.01) with a small effect size (Hedges’ *g* of 0.29 and 0.22, respectively). 

## 4. Discussion

This study focused on examining the FACET-OF-PPC’s factorial validity, internal consistency, and sensitivity to change in order to add to the validation of the first multidimensional outcome measure for pediatric palliative care patients affected by congenital and neurological disorders with severe neurological impairment in Germany. 

The model tested here was developed in a prior study with a comparable patient sample [8]. However, in this study with a new independent patient sample, the model fit was not as good as expected. Results from the confirmatory factor analyses of parent and professional caregiver data, unfortunately, did not quite achieve the cut-off scores suggested by Schreiber [10]. We think that this might be due to the extremely sensitive study population, i.e., pediatric palliative care patients, and especially those with congenital and neurological disorders affected by severe neurological impairment constitute a highly heterogeneous patient group whose health status can change rapidly [2,16,17,18]. These fluctuations in health status, in turn, are likely to influence all aspects of the patient’s and the family members’ lives. Thus, constructing a questionnaire whose factor structure is well replicable with repeated analysis of data from another sample might be unattainable. A factor structure that might have worked well with one sample might not be replicable in another sample, even if it theoretically included the same study participants that were just sampled at a different point in time. We, therefore, argue that the discovered model fit can be seen as appropriate given the characteristics of the patient population in question. Furthermore, Carrozzino et al. [19] have recently pointed out that outcome measures with even suboptimal psychometric criteria may be clinically useful or may even be the only ones feasible. Nevertheless, determining the instrument’s performance in different patient sub-groups might be crucial to further substantiate the properties of the FACETS-OF-PPC.

Over and above, the FACETS-OF-PPC is indeed able to display change over time. Analyses showed significant improvements on the scales “normalcy” and “social support” for parent data and on “child’s social participation”, “normalcy”, and “social support” for professional caregiver data. We hypothesize that changes on the other scales could not be found as the mean amount of time between the two points of assessment for inpatient settings was only 8.9 days for parents and 10.17 days for professional caregivers and 74.5 days for parents and 63.7 days for professional caregivers in the outpatient setting. We cannot expect to see changes in the scales of “coping with the disease” and “caregiver’s competencies” in such a short amount of time. Especially coping with the disease is an extremely sensitive psychological process that takes a considerable amount of time to evolve [5]. Additionally, the development of competencies, for example, the ability to independently perform measures to alleviate the child’s symptoms, does not take place overnight. It is a delicate process requiring a lot of practice [20] and might be unattainable for some families due to the nature of the child’s condition and lack of available resources besides others. 

A similar picture can be found when looking at the change in children’s symptoms according to parents and professional caregivers. Significant improvements can only be found in the parents’ ratings of their child’s sleep problems. Even slight improvements in the child’s sleep may subjectively be seen as a significant change by parents, as many studies have shown that the child’s poor sleep constitutes an enormous strain on the parents’ own health [21,22,23]. Professional caregivers may be able to rate the child’s sleep more objectively. This might explain why the ratings of professional caregivers did not vary significantly between the two time points. Generally, treating the complex symptoms experiences by these children constitutes an enormous challenge [16,24], which might explain why significant changes cannot be displayed in such a short period of time. When inspecting the mean of all symptoms, an overall significant improvement can be found for parent and professional caregiver ratings. The tendency of the individual symptom ratings shown in Figure 1 also suggests that overall, all symptoms tend to improve during the care period. Again, the assessment period may just have been too short to observe more significant changes. 

Furthermore, parents and professional caregivers rated the FACETS-OF-PPC to not be too long while being comprehensible and concerning perceived relevant topics. This adds to the value of the instrument by addressing all relevant aspects of this heterogenous patient population in a comprehensible language and not being too long to use in practice and research. Additionally, the model fits and reliability estimates were very similar for parents and professional caregivers. This underlines the fact that this outcome measure is suitable for both parents and professional caregivers.

Overall, the FACETS-OF-PPC appears to be a promising first multidimensional outcome tool for children with congenital and neurological disorders who are affected by severe neurological impairments. Even though official cut-off scores for the evaluation of its’ model fit could not quite be achieved, the psychometric properties nevertheless suggest that the questionnaire is suitable for the said study population. The FACETS-OF-PPC’s scales show appropriate internal consistency, and the questionnaire is able to demonstrate changes in the assessed scales and symptoms. It is questionable whether any other questionnaire developed for this very heterogeneous patient group would achieve better model fits given the highly participatory and intensive approach taken in the initial development of the questionnaire [8]. Thus, it can be concluded that the results found by Pelke et al. 2021 and those of the current study demonstrate the FACETS-OF-PPC’s validity. Its development complies with the recommendations of the EAPC White Paper on outcome measurement in palliative care [1].

### Strengths/Limitations

By being able to include numerous inpatient and outpatient pediatric palliative care institutions spread throughout Germany, we were able to validate the FACETS-OF-PPC independently of the geographic region and the care setting.

The issue of gate-keeping, meaning that study participants might not have been selected purely based on the inclusion and exclusion criteria but also based on the subjective assessment of whether or not a family is fit to participate in the study, is a common issue in the sensitive setting of palliative care and must always be considered as a limitation [25]. Additionally, the dropout of study participants from the first to the second assessment reduced the sample size that could be included in the analyses. A larger sample might have provided greater statistical power of the analyses and, therefore, clearer results.

## 5. Conclusions

The current study resulted in a further assessment demonstrating adequate psychometric properties of the first multidimensional outcome measure for pediatric palliative care patients with severe neurological impairment, the Family-Centered Multidimensional Outcome Measure for Pediatric Palliative Care, the FACETS-OF-PPC. It provides psychometric data on the questionnaire’s factorial validity, internal consistency, and sensitivity to change. This instrument can be used for clinical and research settings for the aforementioned patient population in Germany.

## Figures and Tables

**Figure 1 children-08-00905-f001:**
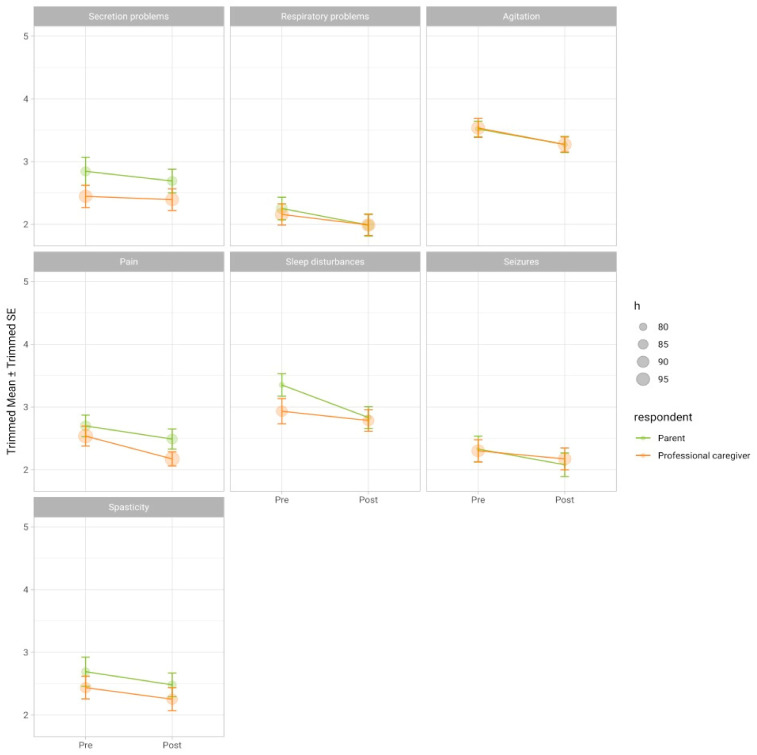
Trimmed mean and standard deviation of the individual symptoms assessed through parents and professional caregivers at admission (Pre) and discharge from an inpatient setting or two months after admission to an outpatient setting (Post); *h* constitutes the sample size after trimming.

**Table 1 children-08-00905-t001:** (**a**) Characteristics of parents and their children (*N* = 227 ^a^). (**b**) Characteristics of professional caregivers (*N* = 238).

(a)	
Parents	
Study participants; *n* (%)	
Mother	172 (75.8%)
Father	19 (8.4%)
Both parents	21 (9.3%)
Other ^b^	11 (4.8%)
Missing	4 (1.8%)
Parents’ age in years; *M* (SD)	
Mothers	41.9 (8.6)
Fathers	44.7 (8.8)
Mothers’ nationalities; *n* (%)	
German	190 (83.7%)
Other	23 (10.1%)
Missing	14 (6.2%)
Fathers’ nationalities; *n* (%)	
German	176 (77.53%)
Others	25 (11.0%)
Missing	26 (11.5%)
**Children**	
Child’s sex	
Male	114 (50.2%)
Female	109 (48.0%)
Missing	4 (1.8%)
Child’s age in years; *M* (SD)	10.9 (6.9)
Child’s nationality; *n* (%)	
German	208 (91.6%)
Other	13 (5.7%)
Missing	6 (2.6%)
Child’s diagnosis ^c^	
E00-E90 Endocrine, nutritional, and metabolic diseases	43 (18.9%)
G00-G99 Diseases of the nervous system	73 (32.2%)
P00-P96 Certain conditions originating in the perinatal period	18 (7.9%)
Q00-Q99 Congenital malformations, deformations, and chromosomal abnormalities	69 (30.4%)
D50-D89 Diseases of the blood and blood-forming organs and certain disorders involving the immune mechanism	1 (0.4%)
Not properly indicated	20 (8.8%)
Missing	3 (1.3%)
Duration of palliative care in years	
0–0.5	41 (18.1%)
>0.5–1	14 (6.2%)
>1–2	19 (8.4%)
>2–5	52 (22.9%)
>5–10	40 (17.6%)
>10	14 (6.2%)
Missing	47 (20.7%)
Sample	
Inpatient	159 (70.0%)
Outpatient	68 (30.0%)
(**b**)	
Sex; *n* (%)	
Male	27 (11.3%)
Female	202 (84.9%)
Missing	9 (3.8%)
Age in years; *M* (SD)	41.5 (12.5)
Work experience in years ^d^; *n* (%)	
0–1	41 (17.2%)
>1–2	27 (11.3%)
>2–5	65 (27.3%)
>5–10	46 (19.3%)
>10	30 (12.6%)
Not indicated	2 (0.8%)
Missing	27 (11.3%)
Work setting ^e^; *n* (%)	
Pediatric Palliative Care Unit	39 (16.4%)
Children’s Hospice	112 (47.1%)
Specialized pediatric palliative home care	77 (32.4%)
Other ^f^	7 (2.9%)
Missing	3 (1.2%)
Profession ^e^; *n* (%)	
Physician	52 (22.5%)
Nurse	184 (79.7%)
Psychologist/social worker	5 (2.2%)
Grief counselor	2 (0.9%)
Other ^g^	6 (2.6%)

^a^ The N included in analyses varies between 180 and 224 due to missing values. ^b^
*n* = 2: care facility, *n* = 1: foster parents, *n* = 7: foster mother, *n* = 1: foster father. ^c^ Indicated by parents and professional caregivers, summarized according to ICD-10 categories, all children suffer severe neurological impairments. ^d^ Within pediatric palliative care. ^e^ percentages exceed 100, as several work settings/professions may be applicable. ^f^
*n* = 2: Outpatient children’s hospice service, *n* = 2: Intensive care unit, *n* = 1: Pedagogical team, *n* = 2: Oncology unit. ^g^
*n* = 1: family office, *n* = 2: team leader, *n* = 2: pedagogical field (siblings), *n* = 1: physical therapy.

**Table 2 children-08-00905-t002:** Descriptive results of the analysis of the evaluation questionnaire.

		*N* ^a^	Mean	SD
Parents	How would you rate the length of the questionnaire? ^b^	212	3.00	0.44
	How comprehensible is the questionnaire? ^c^	216	1.61	0.58
	How relevant are the included items for pediatric palliative care? ^d^	212	1.73	0.67
Professional caregivers	How would you rate the length of the questionnaire? ^b^	223	3.13	0.46
	How comprehensible is the questionnaire? ^c^	222	1.82	0.57
	How relevant are the included items for pediatric palliative care? ^d^	219	1.78	0.62

^a^ N varies due to missing values. ^b^ scale ranges from 1 (far too short) to 5 (far too long). ^c^ Scale ranges from 1 (very comprehensible) to 5 (very incomprehensible). ^d^ Scale ranges from 1 (all items are relevant) to 4 (no item is relevant).

**Table 3 children-08-00905-t003:** Model fits of the confirmatory factor analyses for parent and professional caregiver data for both time points.

	Data	Time Point	*n*	*X2/df*	CFI	TLI	SRMR	RMSEA
Original model ^a^	Parents	Pre	227	2.33	0.73	0.70	0.09	0.08
		Post	165	2.00	0.76	0.73	0.09	0.08
	Professional caregivers	Pre	236	2.71	0.75	0.72	0.11	0.09
		post	190	2.44	0.77	0.74	0.10	0.09
Revised model 1 ^b^	Parents	Pre	227	2.57	0.78	0.75	0.09	0.08
		Post	165	2.39	0.78	0.75	0.09	0.09
	Professional caregivers	Pre	236	2.81	0.82	0.80	0.08	0.09
		post	190	2.46	0.84	0.82	0.09	0.09
Revised model 2 ^c^	Parents	Pre	227	2.39	0.89	0.86	0.07	0.08
		Post	165	2.42	0.86	0.83	0.08	0.09
	Professional caregivers	Pre	236	3.70	0.84	0.80	0.09	0.11
		post	190	2.12	0.92	0.91	0.08	0.08

Note. Values shall ideally achieve the following criteria: X^2^/df < 3, better < 2; CFI > 0.95; TLI > 0.95; SRMR < 0.08; RMSEA < 0.05; ^a^ model as described in Pelke et al., 2021 (see Appendix A). ^b^ model as shown in Appendix A without items B1–B8 assessing symptoms. ^c^ model as shown in Table 4, after iteratively deleting items with poor fit.

**Table 4 children-08-00905-t004:** Questionnaire scales and their respective items.

Scale	Items
Child’s social participation	My child took part in social life according to his/her abilities.
	I have ideas on how to keep my child occupied in daily life.
	Besides his/her limitations, my child also has abilities.
Normalcy	I had time to do the things that make me happy.
	I had time to myself.
	Despite my child’s illness, I was able to maintain social contacts.
	My everyday life was predictable.
	A normal family life was possible for us.
Social support	I was alone in dealing with my child’s illness.
	I was alone with my grief.
	I could talk openly about my child’s illness in my social environment.
Coping with the disease	I despair at the question of why my child is affected.
	I can accept my child’s illness.
	I feel guilty for my child’s illness.
Caregiver’s competencies	I am prepared for my child’s crises.
	If necessary, I am able to independently take measures to alleviate my child’s symptoms.
	I have a clear idea of what should be done for my child in a medical emergency.

**Table 5 children-08-00905-t005:** McDonald’s *ω* for both questionnaire versions and both time points.

	Parents		Professional Caregivers	
	Pre	Post	Pre	Post
Child’s social participation	0.58	0.64	0.67	0.79
Normalcy	0.88	0.88	0.89	0.89
Social support	0.69	0.52	0.65	0.78
Coping with the disease	0.7	0.74	0.67	0.74
Caregivers’ competencies	0.76	0.77	0.84	0.76

**Table 6 children-08-00905-t006:** Results of the robust ANOVAs.

Variable	*F*	*df* _1_	*df* _2_	*p*	*p* _adj_	*n*	*h*
Parents							
Secretion problems	0.93	1	83	0.337	0.385	140	84
Respiratory problems	4.78	1	86	0.031	0.084	143	87
Agitation	4.94	1	76	0.029	0.084	127	77
Pain	2.26	1	85	0.136	0.182	142	86
Sleep disturbances	12.13	1	76	<0.001	0.007	127	77
Seizures	4.23	1	75	0.043	0.086	126	76
Spasticity	2.94	1	80	0.09	0.144	133	81
Professional Caregivers							
Secretion problems	0.15	1	93	0.696	0.696	156	94
Respiratory problems	1.68	1	94	0.197	0.395	157	95
Agitation	4.35	1	94	0.04	0.158	157	95
Pain	6.18	1	98	0.015	0.117	163	99
Sleep disturbances	0.61	1	87	0.436	0.499	144	88
Seizures	0.94	1	92	0.334	0.499	155	93
Spasticity	2.07	1	86	0.154	0.395	143	87

Note. *df*—degrees of freedom; *p*_adj_—*p* value after false discovery rate correction; *n*—sample size; *h*—sample size after trimming.

## Data Availability

The corresponding datasets of this study are available from the corresponding author on reasonable request.

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
