# Peer review of "Validation of the FACETS-OF-PPC as an Outcome Measure for Children with Severe Neurological Impairment and Their Families—A Multicenter Prospective Longitudinal Study"

_children, 2021, doi:10.3390/children8100905_

Round 1
Reviewer 1 Report
This is an outstanding study designed to test the psychometric properties of an
outcome measure (FACETS-OF-PPC) for children with severed neurological impairment and their
families. Per above checklist study design, sample size, data analytic plan and
discussion of results are an important contribution to the field. Although promising,
the authors were not able to find a model that fit the data because of the
heterogeneity of the study sample, a problem in the field especially when studying
children with rare diseases. So my only recommendation might be to temper the
conclusion that the questionnaire is suitable for the study population. However,
I do concur that this questionnaire and that its development complies with the recommendation of the EAPC on outcome
measurement in palliative care.
Reviewer 2 Report
Thank you for the opportunity to review this manuscript, entitled Validation of the FACETS-OF-PPC as an outcome measure for children with severe neurological impairment and their families – A multicenter prospective longitudinal study, which presents an evaluation of the psychometric properties of the FACETS-OF-PPC scale, a measure of multi-dimensional outcomes of children with severe neurological impairment. This manuscript represents an important step toward the development of an instrument that validly and reliably measures multi-dimensional outcomes in children with severe neurological impairment over time. However, several methodological improvements could strengthen the overall value and impact of this manuscript.
Abstract:
Overall, the abstract is concise and adequately summarizes the study. One minor suggestion is in line 23 “the FACETS-OF-PCC has moderate to good internal consistency…”, the authors may consider making this statement more consistent with the findings of the study. In the Discussion, the authors conclude that “The FACETS-OF-PPC scales show appropriate internal consistency” (line 238), which may be more in line with the study’s findings.
Introduction:
Lines 32 to 39: The literature review seems incomplete. The authors might consider referencing recent studies that have been conducted on outcomes in children with severe neurological impairment, such as:
- Feinstein JA, Feudtner C, Blackmer AB, Valuck RJ, Fairclough DL, Holstein J, et al. Parent-Reported Symptoms and Medications Used Among Children with Severe Neurological Impairment. JAMA Netw Open. 2020;3(12):e2029082.
- Bogetz JF, Lemmon ME. Pediatric Palliative Care for Children with Severe Neurological Impairment and Their Families. J Pain Symptom Manage. 2021;62(3):662-7.
- Hauer J, Houtrow A. Pain assessment and treatment in children with significant impairment of the central nervous system. Pediatr. 2017;139(6):e20171002. **references several tools used for pain assessment in children with SNI
Line 37 to 38: Please expand on this point about why a multi-dimensional tool for this population of children with serious neurological impairment is necessary, and why other measures developed for other clinical populations are not appropriate. Some of this is explained in the authors’ previous paper (Pelke, 2021) and might be restated here.
Lines 40 to 52: The authors might consider moving this information about the tool to the Methods section (perhaps combine with the section starting at line 91) so that the focus of the Introduction can be on the gaps in the literature and significance of what this paper adds to the literature.
Line 50: “The first steps towards its validation in Germany…” Please add citation here.
Lines 51 and 56 (and elsewhere in the Methods and Discussion): The authors might consider replacing the phrases “conclude the validation” or “finalizing the FACETS-OF-PPC’s validation” with something to the effect of “a further assessment of the psychometric properties of the FACETS-OF-PPC for use in the population of children with serious neurological impairment in Germany” since the authors are examining the reliability and sensitivity, and not just the validity, of the tool for use in this population of children (and, presumably, could be further evaluated for use in other similar populations of children, so “final” or “to conclude” seems inaccurate).
Line 60: The authors might consider using the phrase “measure change,” rather than “depict change,” as depicting change sounds more descriptive.
Methods
In general, this section should contain information about whether or not participants were compensated for their participation and whether or not ethics/research review board approval was obtained for this study.
Line 64: See comment above about use of the term “validation” (Lines 51 and 56 comment).
Line 83 to 84: The information about the data collection time points isn’t clear as written. From this description, it isn’t clear if the same families completed the questionnaires at discharge for inpatient services and again at two months in the outpatient setting, or if different populations of parents completed the inpatient post-assessment versus the outpatient post-assessment.
Lines 91 to 98: It would also be helpful to know over what timeframe the scale items are assessed. It appears from the Appendix that the scale items ask parents to think about how they are currently feeling, so this might be helpful to include in the main text as well.
Line 102: Rather than the term “suitability,” the authors might consider using the term “acceptability,” which is often used in instrument development studies.
Line 122: More information about the analysis for sensitivity to change would be very helpful for the reader – why was this analysis done? What did it hope to show?
In general, given that the authors collected data at two time points, why was a test-retest reliability analysis not completed? It appears that the internal consistency analysis simply analyzed the sub-scales’ internal consistency at two different time points, rather than assessing the stability of parents’ ratings over time.
Results
Line 126: Please state how many children were represented by this sample of parents and professionals. For example, “A total of N = 227 parents of N= ### children were recruited for study participation…”
Line 129: Given the stark differences in number of days between the two time points, it would be very helpful here to state the sample size for the inpatient sample versus the outpatient sample.
It would be helpful to see the results of the ANOVA analysis, as well as the results of the global rating questions in table format, whether in the Results section or in the Appendices.
The authors should consider breaking down the results by the inpatient and outpatient samples for the pre/post analysis, given the stark differences in time between the two assessments for the two different samples, which could potentially affect the stability of results over time.
Discussion
Line 193: Please clarify the term “extremely sensitive study population.” This reviewer is unclear what “sensitive” means in this context.
Line 203: The authors might consider that for future work, testing this instrument in sub-groups of this population of children might be helpful to understand how this instrument performs across different diagnoses.
Lines 214 to 216: “…the development of competencies…is a delicate process requiring a lot of practice.” The authors might consider also including the idea that for some families, they may never feel fully comfortable managing their child’s symptoms independently, no matter the amount of practice. This may be due to the nature of the child’s disease, the support and resources families have (particularly in the home environment), etc.
Lines 242-245: The authors should consider comparing, in more detail, the findings of this study to the findings of their previous study and how they differ, are the same, etc. earlier on in the Discussion section.
The authors might comment in the Discussion section about the interpretation of the results of the parents’ evaluation of the tool (Table 2 results).
The authors might also comment in this section about any differences or similarities between parents’ and professionals’ results, as well as differences/similarities between inpatient and outpatient results.
Lines 247 to 256: Could parents’ health literacy have impacted how they answered the questions in this instrument? While this reviewer is not familiar with the German language, could some parents be less fluent in German, which could impact their results?
Conclusions
Line 258: Similar to the comment earlier, the authors should consider qualifying this statement to say something like “The findings from the current study show adequate psychometric properties for use of the FACETS-OF-PPC instrument in children with severe neurological impairments in Germany,” as, similar to any other instrument, it would need to be tested for use in other populations.
Line 262: Similar to the previous comment, the conclusion that the instrument can be used for clinical and research settings may be overstated – again, it would require further evaluation for use in other countries, settings, disease sub-groups, etc.
Tables
Table 1: Please also include the “n” for missing values for each of the variables (i.e., mothers v. fathers, mothers’ nationalities, fathers’ nationalities). The total for each variable should ideally be equal to the total “N” of the study, whether or not participants answered the questions. As presented, the percentages are confusing, as the denominator is different for each variable.
Table 3: This reviewer appreciates the footnote in this table, which provides the reference criteria for the X2, CFI, etc.
The authors should strongly consider including a table, whether within the main text or in the Appendices, which contains all original scale items arranged by domains, and depicts which items were deleted at which stage (similar to a PRISMA diagram, perhaps). While the original scale items are included in the Appendix, it is difficult to follow since the ordering of the items are different than in the main text, and the sub-scale/domain names are different than what is presented in the main text.
Round 2
Reviewer 2 Report
Thanks to the authors for their very thoughtful revisions and responses to the reviewer comments. One remaining suggestion is to remove the footnote in Table 1a "The N included in analyses varies between 211 and 231 due to missing values" since the N was adjusted to include missing values. Otherwise, this manuscript appears to be significantly improved.
